# Unveiling immune cell response disparities in human primary cancer-associated fibroblasts between two- and three-dimensional cultures

Jian-Ping Yang[1]*, Nikhil Nitin Kulkarni[1], Masashi Yamaji[1], Tsubasa Shiraishi[2], Thang Pham[3], Han Do[3], Nicole Aiello[4], Michael Shaw[5], Toshihiro Nakamura[2], Akiko Abiru[2], Narender R. Gavva[1], Shane R. Horman[1]

1 Takeda Development Center Americas, Inc., San Diego, California, United States of America, 2 Takeda Pharmaceutical Company Ltd, Fujisawa, Kanagawa, Japan, 3 BioTuring, San Diego, California, United States of America, 4 Bristol-Myers Squibb, Princeton, New Jersey, United States of America, 5 Takeda Development Center Americas, Inc., Cambridge, Massachusetts, United States of America

* jpy8888@yahoo.com

**Data Availability Statement:** The SnRNA-seq data supporting this publication are available at https://doi.org/10.6084/m9.figshare.27289656.

## Abstract

Cancer-associated fibroblasts (CAFs) play pivotal roles in solid tumor initiation, growth, and immune evasion. However, the optimal biomimetic modeling conditions remain elusive. In this study, we investigated the effects of 2D and 3D culturing conditions on human primary CAFs integrated into a modular tumor microenvironment (TME). Using single-nucleus RNA sequencing (snRNAseq) and Proteomics' Proximity Extension Assays, we characterized CAF transcriptomic profiles and cytokine levels. Remarkably, when cultured in 2D, CAFs exhibited a myofibroblast (myCAF) subtype, whereas in 3D tumor spheroid cultures, CAFs displayed a more inflammatory (iCAF) pathological state. By integrating single-cell gene expression data with functional interrogations of critical TME-related processes [natural killer (NK)-mediated tumor killing, monocyte migration, and macrophage differentiation], we were able to reconcile form with function. In 3D TME spheroid models, CAFs enhance cancer cell growth and immunologically shield cells from NK cell-mediated cytotoxicity, in striking contrast with their 2D TME counterparts. Notably, 3D CAF-secreted proteins manifest a more immunosuppressive profile by enhancing monocyte transendothelial migration and differentiation into M2-like tumor-associated macrophages (TAMs). Our findings reveal a more immunosuppressive and clinically relevant desmoplastic TME model that can be employed in industrial drug discovery campaigns to expand the cellular target range of chemotherapeutics.

## Introduction

In recent years, the tumor microenvironment (TME) has emerged as a focal point in cancer research, influencing immune responses, metastasis, and therapeutic outcomes [1–3]. Comprising a complex network of components, the TME includes tumor cells, infiltrating immune cells (such as macrophages, dendritic cells, and lymphocytes), cancer-associated stromal cells

**Funding:** The author(s) received no specific funding for this work.

**Competing interests:** The authors have declared that no competing interests exist.

(notably, cancer-associated fibroblasts or CAFs), and other cell types, as well as non-cellular elements, such as the extracellular matrix (ECM) and soluble factors secreted by different cell types. Non-malignant cells within the TME exhibit genomic stability, but remain adaptable. Their transcriptomes and phenotypes are shaped by their interactions with cancer cells and other TME components [4, 5].

CAFs are the main stromal components of solid tumors [6, 7]. Under basal homeostatic conditions, fibroblasts maintain a tissue equilibrium and facilitate essential cell communication. However, when activated by factors secreted from cancer or immune cells, fibroblasts undergo a transformative shift, suggesting a role for CAFs [143, 169]. These activated CAFs exhibit remarkable biological heterogeneity due to their diverse origins, phenotypes, and functional molecules. Even within a single tumor, CAFs may consist of diverse subpopulations, each with distinct functions. Recent investigations of CAF signaling networks have defined subtypes according to function: myofibroblastic CAFs (myCAFs), which are involved in ECM maintenance and wound healing regulation; inflammatory/immune regulatory CAFs (iCAF), characterized by their secretory phenotype and immune cell regulatory activity; and antigen-presenting CAFs (apCAFs), which express MHC class II molecules [8, 9]. Each of these CAF subtypes has a significant influence on the TME, particularly through their properties of immunomodulation. By interacting with cancer and immune components, CAFs contribute to immunosuppression and surveillance escape, orchestrating a delicate balance that affects tumor progression and response to therapy [10].

CAFs actively shape the TME by remodeling the organ's extracellular matrix and releasing an array of cytokines and chemokines that facilitate accessory cells to do the same [11, 12]. Their intricate dance with cancer cells, often described as "partners in crime", fuels tumor growth, angiogenesis, inflammation, and drug resistance [13, 14]. Numerous studies have identified CAFs as viable targets for clinical intervention [15, 16]. Therefore, understanding the dynamic interplay among cancer cells, CAFs, and immune cells is crucial for innovative CAF-targeted drug development [17, 18].

However, several challenges have hindered our progress in the study and development of therapeutic strategies that target CAFs. CAFs exhibit remarkable heterogeneity in their origin and function, which complicates modulation for therapeutic benefits [19, 20]. Additionally, the lack of validated in vitro cell culture systems specifically tailored for studying human CAF biology remains a critical gap in preclinical evaluation of therapeutic efficacy.

Although conventional 2D cell culture methods remain the industry standard for drug discovery owing to their simplicity and scalability, they fail to accurately represent in vivo behavior [21, 22]. Cell-to-substrate contacts in 2D (as opposed to cell-to-cell contacts in 3D) induce altered phenotypes that diverge from the complex in vivo milieu. Traditionally, animal models have been used to assess drug efficacy and toxicity. However, animal tumor growth studies may not fully mimic human tumor biology [23]. In addition, animal models have various issues, such as substantial costs, logistical demands, limited bioavailability, and ethical concerns.

3D culture is a unique technology that enables intricate tumor–immune interactions through the hierarchical organization of TME cell types within a microtumor spheroid [24, 25]. By co-culturing multiple cell types in 3D microtumors (neoplastic epithelial cells, primary patient-derived CAFs, and primary human immune cells), we gained critical insights into cellular dynamics within the TME [26] which may reveal novel therapeutic targets and biomarkers. Given that new medicines no longer require animal testing for U.S. Food and Drug Administration (FDA) approval [27], enhancing preclinical models with greater complexity, such as organoids [28] and organ-in-a-chip systems [29], to mimic the in vivo TME would significantly facilitate the drug development process.

In this study, we meticulously explored various in vitro culture methods for patient-derived human primary CAFs to identify an in vitro CAF model that faithfully mimicked the clinical interactions between CAFs, cancer cells, and immune cells. Armed with single-nucleus RNA sequencing (snRNAseq) and highly sensitive proteomics detection, we characterized transcriptomic profiles of primary pancreatic cancer patient CAFs cultured in 2D and 3D monoculture and co-cultures with pancreatic cancer cells. We showed that different CAF culture conditions have profound effects on antitumor function and polarization of natural killer (NK) cells.

These findings may be applied to the standardization and employment of next-generation ex vivo human tumor models to enhance our knowledge of cancer biology and represent clinically relevant platforms for the evaluation of therapeutic efficacy.

## Material and methods

### Antibodies and reagents

The primary antibodies used for flow cytometry, immunofluorescence, immunohistochemistry, and immunoblotting were CD206 (Cat. no. 321104; BioLegend), CD163 (Cat. no. 333606; BioLegend), α-SMA (Cat. no. ab5694, Abcam,), and GAPDH (Cat#: 5174; Cell Signaling Technology). CD14 microbeads (Cat. no. 130-050-201, Miltenyi Biotec) were used to sort monocytes. Butylated hydroxyanisole (BHA) was purchased from Sigma–Aldrich.

### Cell cultures

Human pancreatic CAF-Stellate Cells were purchased from Neuromics and cultured in CAF growth media (Neuromics). BxPC3_Red_luc cells were cultured in RPMI1640 or Dulbecco's Minimal Essential Medium supplemented with 10% fetal bovine serum.

### SnRNAseq and Olink panel analysis

For 2D cultures, $5x10^5$ human pancreatic CAFs or a mix of $1x10^5$ BxPC3 (RRID:CVCL_XX78) cancer cells and $3x10^5$ pancreatic CAFs per well were cultured in 6-well plates in 2 ml RPMI 1640 with 0.5% fetal bovine serum (FBS) culture media for monoculture and co-culture, respectively. For 3D culture, 2x104 pancreatic CAFs or a mix of a mix of $0.5x10^4$ BxPC3 cells with $1.5x10^4$ pancreatic CAFs per well were seeded in 100 μL of RPMI 1640 with 0.5% FBS culture medium in 96-well round-bottom, ultra-low attachment plates (Corning Inc., New York, USA), centrifuged at 500×g for five minutes then placed in a 37°C incubator at 5% CO2 for 2 days. The cells and spheroids were harvested, washed with DPBS, and frozen for RNA-seq analysis. Supernatants were collected and centrifuged to remove cell debris for Olink analysis. The quality control (QC) for single-nucleus RNA sequencing (snRNAseq) data analysis is provided in S1 Fig.

### Tumor cell growth measurement

$5x10^3$ BxPC3 mCherry-luficerase reporter cells were monocultured or co-cultured with $1x10^4$ CAFs in flat-bottom or ultra-low attachment 96-well plates for 2D and 3D cultures, respectively. The Incucyte live-cell imaging platform (Sartorius) tracked BxPC3 killing over a period of 5 days for both 2D and 3D cultures. At the end of the imaging, the BxPC3 residual cell count was assessed using the Bright-Glo luciferase system (Cat. No. E2620, Promega) on the GloMax plate reader (Promega).

## NK cell cytotoxicity assay

Primary peripheral blood human NK cells were acquired from healthy nonsmoking donors (Cat. no.70036, Stem Cell Technologies). NK cells were cultured in CTS NK Xpander medium (Cat. No. A5019001, ThermoFisher Scientific), supplemented with 5% human AB serum (Cat. No.H4522, Sigma Aldrich), and 200 ng/ml IL-2 (Cat. no. 202-IL-010; R and D Biosystems) for 48 h post thaw to generate activated NK cells.

For the NK cell killing assay, IL2 activated NK cells were cultured with either BxPC3 pancreatic adenocarcinoma cells or co-cultured BxPC3 and CAF cells in both monolayer 2D and spheroid 3D format in 96 well tissue culture plates (Cat. No. 3595 and 4520, Corning), as described in the section on tumor growth measurements. Effector (NK cells) to target (BxPC3) ratios of 5:1, 2.5:1, 1.25:1, and 0.625:1 were used in the NK cell killing assay. The Incucyte live-cell imaging platform (Sartorius) tracked BxPC3 killing over a period of 7 days for both 2D and 3D cultures. At the end of the imaging, the BxPC3 residual cell count was assessed using the Bright-Glo luciferase system (Cat. No. E2620, Promega) on the GloMax plate reader (Promega).

## CAF condition media generation

Human pancreatic CAFs were cultured in the CAF growth medium (Cat. No. CAFM03, Neuromics). A total CAF cell density of $0.335 \times 10^6$ cells/ml was maintained for both monolayer (2D) and spheroid (3D) CAF culture. A total of $1.34 \times 10^6$ cells were cultured in 4 ml reduced serum medium (RPMI-1640, 2 mM glutamine, and 0.5% Fetal Calf Serum) in each well of Aggrewell 800 (Cat. no. 34825; Stem Cell Technologies) for 3D culture with 900 cells seeded per microcavity. The 6-well Aggrewell microcavity plate was briefly centrifuged at $300 \times g$ for 2 min to allow CAF spheroid formation. For 2D CAF culture, $0.668 \times 10^6$ cells were cultured in 2 ml reduced serum medium in each well of a 6 well tissue culture plate (Cat. No. 140675, Nunc Delta, ThermoFisher Scientific). Conditioned supernatant was collected 72 h post CAF culture for both 2D and 3D, filtered through a 40-u cell strainer, and frozen at -80˚C.

## Monocyte purification

Leukocytes were obtained from healthy volunteers. Monocytes were isolated using CD14 negative selection methods by Robosep from StemCell. The purity and recovery of CD14+ monocytes were determined using CD14-FITC antibody labeling (Cat. no. 130-113-708; Miltenyi Biotec), whereas cell viability was determined by propidium iodide (PI) staining and flow cytometry analysis using a BD FACSAria III (BD Biosciences, San Jose, CA, USA).

## Monocyte transendothelial migration assay

Human endothelial cells (HUVECs, RRID:CVCL_2959) were purchased from StemCells and cultured in EBM$^{TM}$-2 Endothelial Cell Growth Basal Medium-2 supplemented with 5% fetal bovine serum (FBS), hydrocortisone, human fibroblast growth factor-beta (hFGF-b), vascular Endothelial Growth Factor (VEGF), R3-Insulin-like growth factor-1 (R3-IGF-1), ascorbic acid, epidermal growth factor (hEGF), and gentamicin/amphotericin-B (GA). IncuCyte Clear-View 96-Well Chemotaxis Cell Migration Plates were coated with 5 mg/mL fibronectin diluted with D-PBS, and 8000 HUVECs per well were seeded into the top chamber of a Transwell and incubated overnight to form a confluent monolayer. Subsequently, monocytes pre-stained with the green CellTracker dye were added to the top chamber. Monocyte migration was monitored by imaging the top and bottom membranes of cells in an incubator. Quantitative data demonstrated that conditioned media from 3D CAFs significantly enhanced monocyte

transendothelial migration compared to that from 2D CAFs. Fluorescent labeling of mono-cytes and chemotaxis plate preparation with HUVEC monolayers. Real-time analysis using the IncuCyte® S3 live-cell imaging system, image analysis, and assessment of the transendothelial migration rates. CCL2 (100 ng/ml) was used as a positive control.

## Macrophage differentiation and polarization

Monocytes were isolated from the leukopaks of healthy non-smoking donors (cat. no. 70500.2, Stem Cell Technologies), using the Easy Sep Human Monocyte Isolation Kit (Cat. 19356 RF, Stem Cell Technologies) on a RoboSep cell isolation automated platform (Stem Cell Technologies), according to the manufacturer's instructions. The purity of the isolated monocytes was verified using flow cytometry (CD14+ CD15 −). Monocytes were cryopreserved in Cryostor CS10 (Cat. no. 210502; Biolife Solutions).

Frozen monocytes were thawed and cultured in Immunocult SF Macrophage Medium (Cat. no. 10961, Stem Cell Technologies). A total of $0.5 \times 10^6$ cells were seeded in each well of 12 well tissue culture treated plate in 500 μL medium. Conditioned medium (500 μL) from the 2D or 3D CAF cultures was added to 50% of the total volume. The effect of CAF-conditioned medium on macrophage differentiation and polarization was assessed 48-h and 7 d after CAF-conditioned medium treatment. On days 2 and 7, cells were harvested using Accutase (Cat. No. 00-4555-56, Thermo Fisher Scientific) and analyzed using flow cytometry.

## Flow cytometry

Macrophages were washed with 1X DPBS and stained with fixable aqua (Cat. No. L34957) at 1:1000 dilution in 1X DPBS for 15 min. The cells were washed with FCS staining buffer (Cat. N0.554647; BD Biosciences), and incubated with an Fc receptor blocker (Cat. No.422302; Bio-Legend) for 10 min. Primary conjugated antibodies (S1 Table) were incubated for 20 min in the dark at room temperature. IgG controls were included in all antibodies (S2 Table). Follow-ing incubation with the primary antibody, cells were washed with FCS buffer. The cells were incubated for 30 min at 4 C in the dark. The cells were then washed twice with FCS buffer (800 g, 2 min). For intracellular CD68 staining, 100 μL fixation/permeabilization solution was added to the cells for 20 min. Next, the cells were washed twice with 1× BD Perm/Wash Buffer. CD68 antibody was added at a 1:100 dilution in Perm/Wash buffer, and the cells were incu-bated at RT for 30 min in the dark. Cells were washed twice in perm/wash buffer and resus-pended in Staining FCS Buffer prior to flow cytometric analysis. The cells were analyzed using an Attune Nxt flow cytometer (ThermoFisher Scientific). The gating strategies employed for the FACS analysis of macrophage phenotypes were as follows: dead cells were excluded from the analysis; monocyte-derived macrophages were gated as M1 (CD45+ CD14 lo/- CD11b + CD80high/+ CD86 high/+), M2a (CD45+ CD14+ CD11b+ CD206 high/+), and M2c (CD45 + CD14+ CD11b+ CD163 high/+).

The panels of macrophage-specific markers and IgG controls for flow cytometry analysis are listed in S1 and S2 Tables respectively.

## MESO ELISA

CAF-secreted protein expression in the cell-conditioned medium was analyzed using the MSD multi-array multiplex platform, according to the manufacturer's instructions. (Meso Scale Dis-covery, Rockville, MD, USA). Briefly, MSD multi-array 96 well plates were coated with a spot-specific linker and capture antibody mix, and incubated with shaking for 60 min at 750 rpm at room temperature. After capture antibody incubation, CAF-conditioned medium samples were added to the plates at 1:2 and 1:10 dilutions along with the standard calibrators provided

in the kit. The Samples and calibrator standards were incubated overnight at 4°C. Following overnight incubation, sulfo-tag-conjugated detection antibodies were added to the samples and incubated at room temperature with shaking at 750 rpm for 60 minutes. The plate was washed thrice with MSD buffer between each step. The readout was performed after the addition of MSD Gold Read Buffer. ECL was detected using an MSD Meso Sector S600 MM reader and the data were analyzed using the Discovery Workbench.

### Ethics statement

This article does not contain any studies involving human participants performed by any of the authors.

## Results

### CAFs cultured in 2D or 3D yield distinct CAF subtypes

Pancreatic ductal adenocarcinoma (PDAC) is characterized by a significant amount of desmoplasia, accounting for up to 90% of the total tumor volume [30, 31]. The microenvironment of pancreatic ductal adenocarcinoma (PDAC) is distinguished by a dense, fibrotic stroma, with cancer-associated fibroblasts (CAFs) serving as pivotal contributors [32]. Recently, researchers have classified various subtypes of CAFs in PDAC patients according to their transcriptome signatures [33, 34]. To understand how the growth of PDAC patient-derived primary CAF under 2D and 3D conditions influences the transcriptomic profiles of both CAFs and cancer cells, we used single-nucleus RNA sequencing (snRNA-seq) to detect highly sensitive and cell-specific gene expression. Simultaneously, we collected the culture supernatants and assessed the expression of a panel of immuno-oncology and inflammation markers by using a proximity extension assay (Fig 1A).

The human pancreatic cancer cell line BxPC3 and primary patient-derived pancreatic CAFs were used for TME modeling studies. Clustering analysis of snRNA-seq data confirmed that CAFs and cancer cells exhibited distinct transcriptional profiles based on the dimensionality of their growth (Fig 1B). Despite the lack of a consensus CAF molecular signature, several biomarkers have been used to distinguish different CAF subtypes in PDAC patient samples, including ACTA2, TAGLN, MYL9, TPM1, TPM2, HOPX, and POSTN for myofibroblastic CAFs (myCAFs), and CXCL8, CXCL2, PDGFRA, IL6, CFD, DPT, AGTR1, HAS1, CXCL1, CCL2, IL8, and LMNA for inflammatory CAFs (iCAFs) [35]. Interestingly, CAFs cultured in 2D, whether in monoculture or co-culture with tumor cells, exhibited elevated expression levels of myCAF-associated marker genes (Fig 1C). Conversely, CAFs cultured in 3D monoculture showed increased expression of marker genes associated with iCAFs, including CCL2, CXCL8, and PDGFRA, but had indistinguishable IL6 levels compared to CAFs cultured in 2D and 3D. Interestingly, iCAF signatures were subdued on CAFs when co-cultured with cancer cells in a 3D culture. We did not observe any antigen-presenting CAFs (apCAFs) based on the expression markers H2-Aa, H2-Ab1, CD74, CD239, or CD321 in any 2D or 3D CAF model.

The transcript levels detected by snRNA-seq and the secreted protein levels produced by CAFs under varying culture conditions showed significant compositional differences (Fig 2D). For the most part, gene transcript levels measured by snRNA-seq and protein levels measured by Olink for the same gene were strongly correlated, with a few exceptions (Fig 2B), implying that transcriptomic values reliably represented translated protein levels. Notably, 3D-cultured CAFs exhibited elevated transcript and protein expression levels of CCL7, CCL8, IL24, LIF, VEGFA, CXCL8, TNFRSF11B, and MMP10 compared to their 2D counterparts. These genes play critical roles in biological pathways related to inflammation, immune responses, tissue

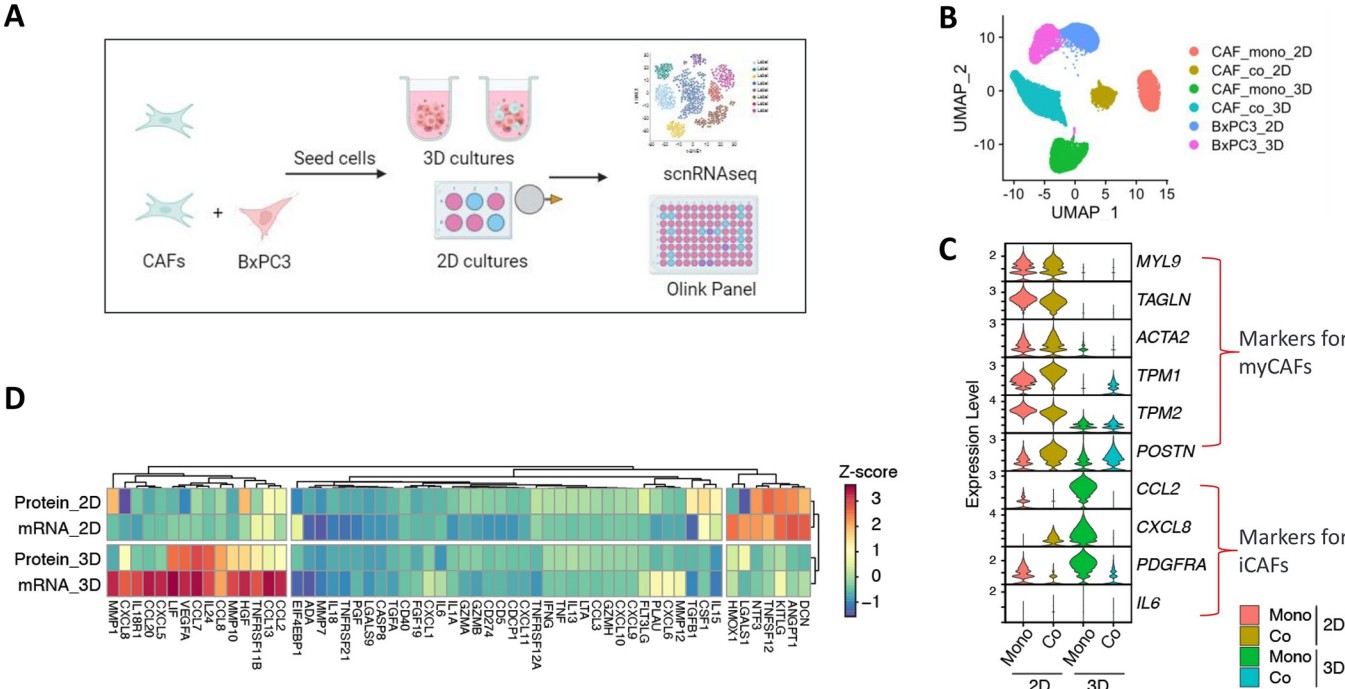

**Fig 1. Characterization of CAFs using scnRNAseq and proteomic profiling.** A) Workflow for sample collection. Patient-derived primary pancreatic CAFs and pancreatic cancer cells BxPC3 were cultured in 2D and 3D. Cells were collected for scnRNAseq and the culture supernatants were collected for protein measurements 72 h post seeding. B) Cell clusters identified by scnRNAseq. C) Comparison of biomarker gene expression of myCAF and iCAF on CAFs in 2D and 3D monoculture and co-cultured with cancer cells. D) Heat map comparing transcriptomic levels of genes detected by scnRNAseq and secreted protein expression levels detected by a proximity extension assay in the supernatants of CAF supernatants from 2D and 3D monocultures.

repair, and angiogenesis and emphasize the distinct gene expression differences of CAFs cultured in 2D versus 3D.

## CAFs and cancer cells grown in 2D or 3D co-cultures reveal divergent transcriptomic profiles and matrix metalloproteinases (MMPs) signaling networks

We observed distinct transcriptomic profiles in cancer cells and CAFs when co-cultured with CAFs under 2D and 3D conditions, as revealed by the snRNAseq data analysis (Fig 2A). Notably, CAFs in 2D monoculture showed elevated expression of TGFB2, NREP, MEGF6, and POU2F2, whereas CAFs in 2D co-culture showed elevated expression of PTN, ALDH1A1, PGS10, MFAP4, RGCC, LHB, ALDH2 and LMCD1. CAFs grown in 3D monoculture showed elevated expression of CCL7, LC16A6, NR4A2, MMP3, BDKRB2, CHI3L1, TWIST1, GK, HGH, IL24, and TRPA1, whereas CAFs in 3D co-culture showed elevated expression of FAM2168, LINC01705, MMP11, and MMP9 (Figs 2B and 3C).

## CAFs in 3D co-cultures promoted tumor growth

The role of CAFs in promoting tumor progression has been extensively reported in different types of solid organ cancers [36, 37]. Because CAFs cultured in 2D or 3D displayed different transcriptomic/proteomic profiles of CAF subtypes, we next assessed how these subtypes affected cancer cell growth. The BxPC3 PDAC cell line was engineered to express both a red fluorescent protein and luciferase reporter, allowing us to monitor cancer cell growth using both high-content imaging (Incucyte) and luciferase assays (Brite-Glo) (Fig 2A). Fig 3B and 3C display images

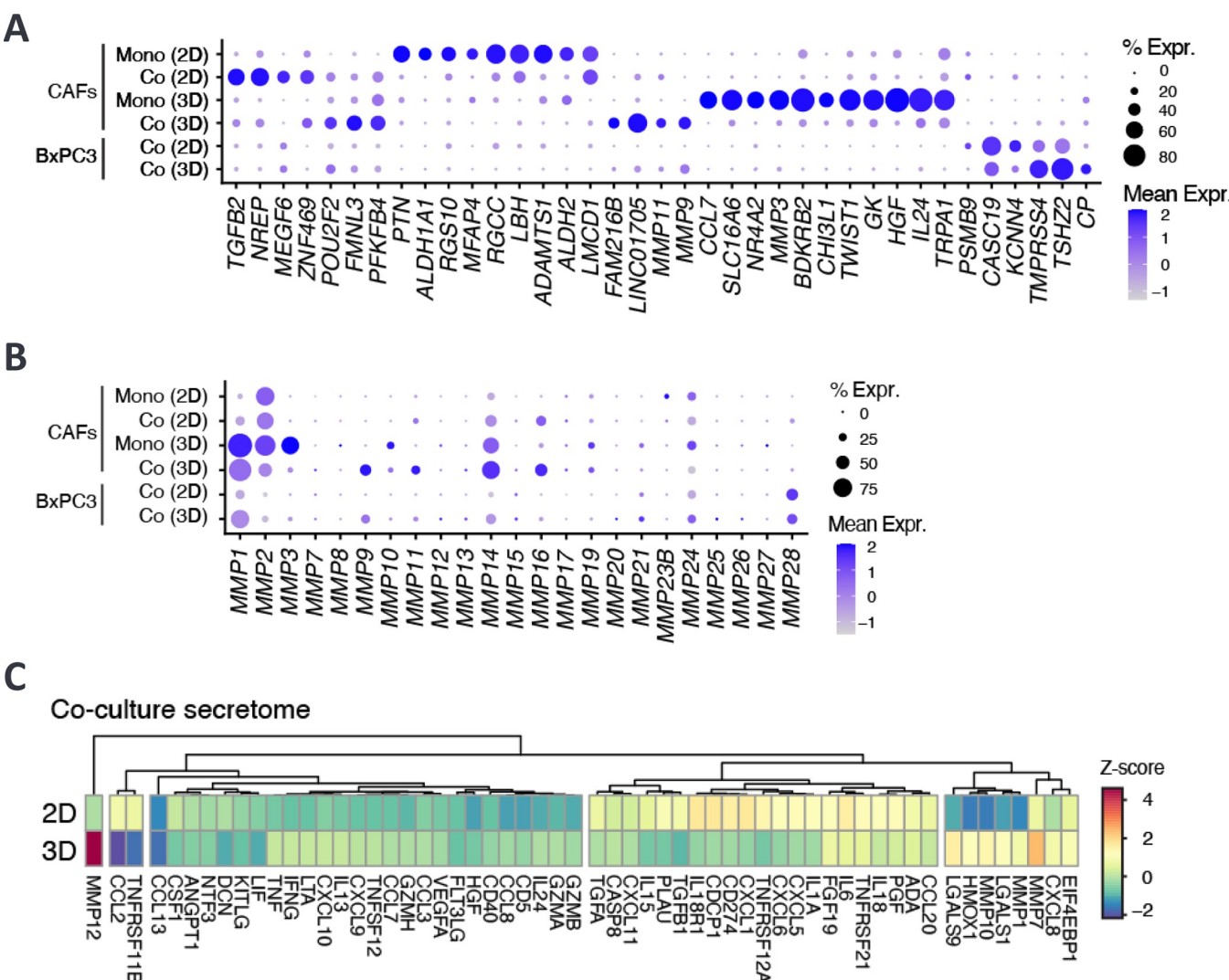

**Fig 2. Culture conditions significantly affect gene expression profiles of CAFs and cancer cells.** A) Differentiated gene expression of CAFs and cancer cells in 2D and 3D monocultures and co-cultures based on scnRNAseq data. B) Transcriptomic expression levels of MMPs on CAFs and cancer cells in 2D and 3D monocultures and BxPC3 co-cultures. C) Comparison of the secreted protein expression levels in the supernatants of 2D and 3D cancer and CAF co-cultures.

of red fluorescent cancer cells in both monoculture and co-culture with CAFs in 2D and 3D and reveal significant growth differences in 3D compared to 2D. We tracked the number of cancer cells in 2D cultures over time and noticed that co-culture with CAFs in a 2D environment did not significantly affect cancer cell growth, as evidenced by both the imaging analysis and end-point luciferase assays (Fig 3D and 3E). In the 3D setting, we assessed the overall fluorescence intensity of tumor spheroids with or without CAFs and observed that co-culture with CAFs in 3D yielded a substantial increase in cancer cell growth compared to cancer cell monoculture in 3D (Fig 3F). Quantification of cancer cell growth by luciferase assay confirmed that co-culture with CAFs in 3D culture resulted in significantly greater tumor spheroid growth than 3D cancer cell monoculture (Fig 3G). These data revealed protumorigenic properties of CAFs grown in 3D co-cultures that were not observed when cultured in 2D formats.

CAFs play a pivotal role in solid tumor growth and maintenance by actively remodeling the stromal ECM through the secretion of matrix metalloproteinases (MMPs) [38, 39]. This

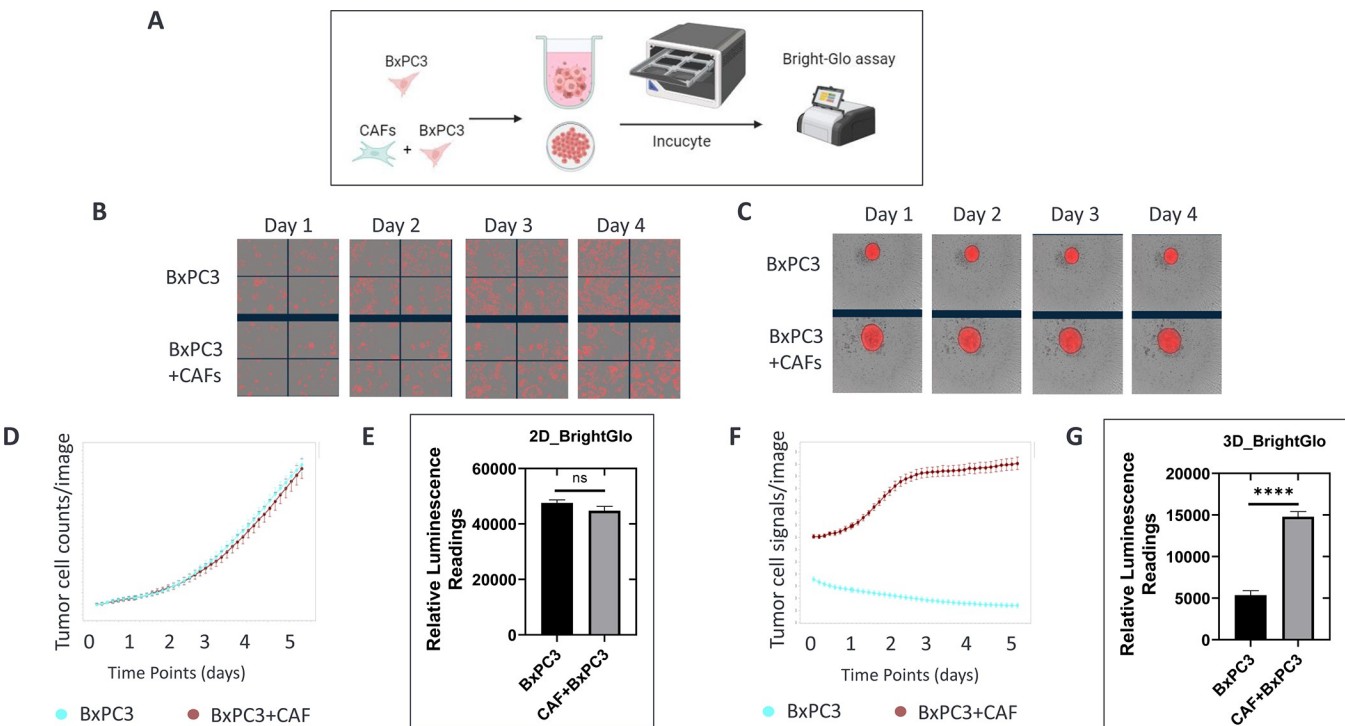

**Fig 3. CAFs in 3D co-cultures promote cancer cell growth.** A) BxPC3 pancreatic cancer cells were monocultured or co-cultured with human pancreatic CAFs in 2D and 3D. Cancer cell growth was monitored by Incucyte imaging for 5 days and end-point luciferase assay at day 5. B and C) Cancer cell images of 2D (B) and 3D (C) at days 1, 2, 3, and 4. D and F) Cancer cell growth measured by Incucyte: Cancer cell counts (D) and fluorescence intensity of spheroids (F) are plotted as y-axis. E and G) Cancer cell viability for 2D (E) and 3D (G) was measured with Bright-Glo reagents at day 5. Relative luminance units were expressed as y-axis.

remodeling process provides essential structural support for tumor invasion and neoangiogenesis. We dissected the impact of culture dimensions on MMP expression by using transcriptomic and proteomic data. Notably, CAFs grown in the 3D monoculture exhibited elevated expression levels of MMP1, MMP2, MMP3, MMP10, and MMP14 compared to their 2D counterparts (Fig 3B). The transcriptomic data aligned with the equivalent protein expression levels detected in the supernatants from 3D co-cultures, which exhibited increased levels of MMP1, MMP7, MMP9, MMP10, and MMP12 compared to 2D co-cultures (Fig 3C). Interestingly, the elevated expression of MMP1, MMP2, and MMP3 observed in 3D CAF monocultures was downregulated in 3D co-cultures, indicating a tumor-mediated suppressive effect on these genes. Intriguingly, MMP9 expression was specifically upregulated in both tumor cells and CAFs only in the 3D co-cultures [40].

Proteomic analyses revealed similarly distinct expression levels of secreted proteins in the supernatants of CAFs and cancer cell co-cultures (Fig 3C). Notably, several cytokines of the CXCL family displayed higher expression levels in 2D than in 3D co-cultures. These proteins function in immune cell recruitment, tumor biology, and disease pathogenesis.

## CAFs grown in 3D tumor spheroids suppressed NK cell anti-tumor function

Natural killer (NK) cells play an essential role in tumor immune surveillance, exhibiting an innate ability to kill tumor cells without prior priming [41, 42]. To understand the functional implications of divergent gene and protein expression in CAFs grown in different formats, we

explored the effect of CAFs on tumor sensitivity to activated human primary NK cells. We used the PDAC TME model, BxPC3 cells that express a red fluorescent protein and luciferase reporter, to enable both high-content kinetic imaging (Incucyte) and tumor viability (Brite-Glo) assays to monitor cancer cell growth (Fig 4A). We assessed NK cell-mediated antitumor

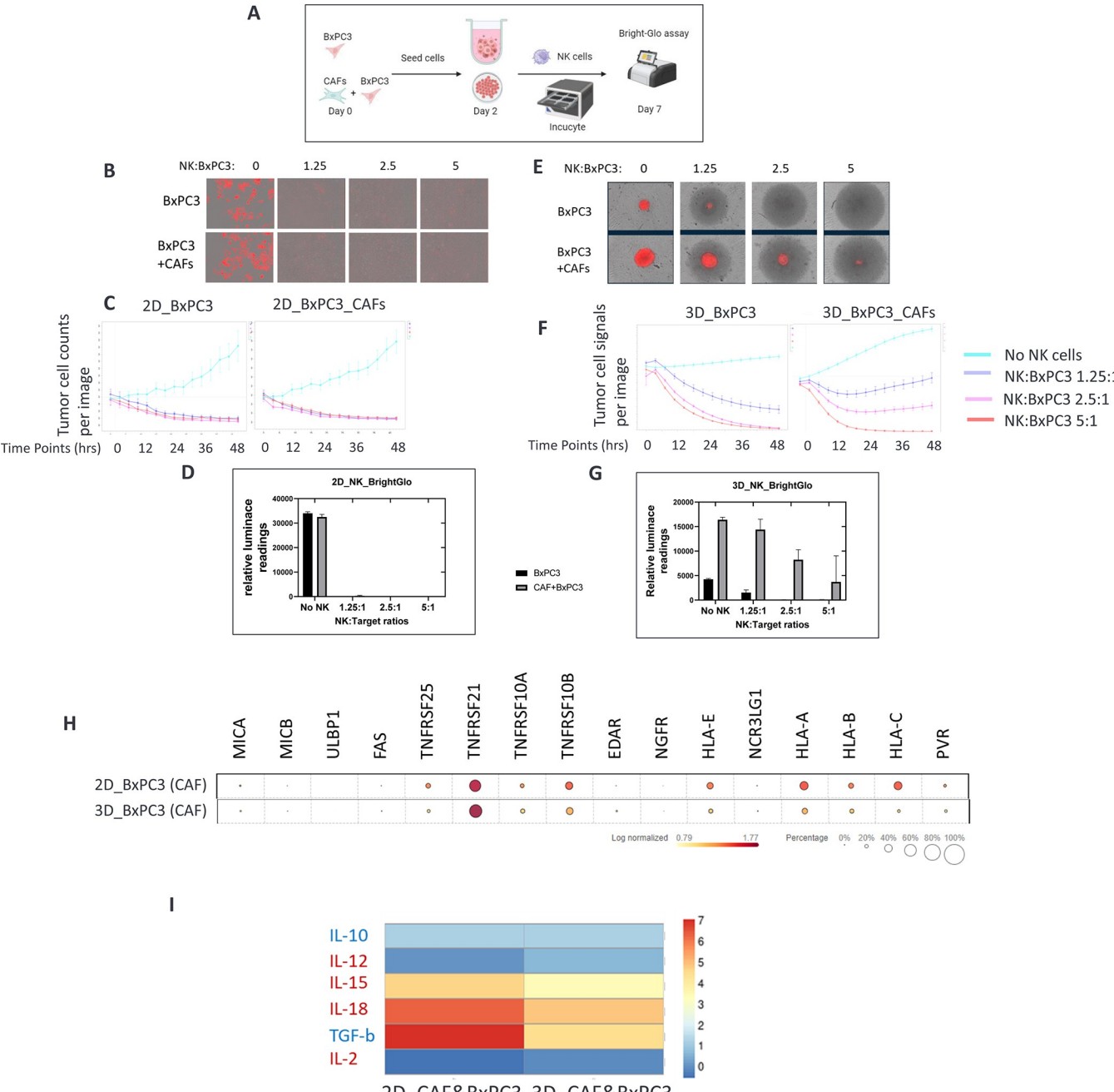

**Fig 4. CAFs in 3D co-culture protect against NK killing.** A) CAFs and cancer cells were seeded in 2D and 3D for 24 h and activated human primary NK cells were added to cell cultures. Cancer cell growth was monitored by Incucyte imaging for 5 days and end-point luciferase assay at day 5. B and C) Images of cancer cells with 1:0, 1:1.25, 1:2.5, and 1:5 ratios of target to effect cells in 2D and 3D. Cancer cell counts (D) and fluorescence intensity of spheroids (F) are plotted on y-axis. E and G) Cancer cell viability for 2D (E) and 3D (G) was measured with Bright-Glo reagents post incubation with NK cells for 5 days. Relative luminescance units are shown on y-axis. H) Transcriptomic analysis of ligands related to NK killing on cancer cells in 2D and 3D co-cultures. I) Heat map of cytokines that are related to NK killing activity in the supernatants of 2D and 3D co-cultures.

function at varying effector-to-target (E:T) ratios to identify dose-responsive effects. High-content kinetic imaging studies revealed substantial NK-mediated tumor cell killing at all E:T ratios in 2D growth formats (Fig 4B). The data from both Incucyte imaging and luciferase end-point assays were consistent with 2D cell cultures (Fig 4C and 4D), and activated human primary NK cells effectively eradicated cancer cells regardless of the presence or absence of CAFs. However, these NK antitumor effects were significantly suppressed when cancer cells were grown in 3D and in the presence of CAFs (Fig 4E). We observed a modest reduction in the fluorescence intensity of cancer cells when the spheroids in the 3D co-culture were treated with activated NK cells compared to that of the 3D tumor monoculture (Fig 4F). This observation was confirmed by the end-point luciferase results (Fig 4G), which suggested that CAFs might confer protection to tumor cells from NK cell killing in 3D culture settings.

NK cell activation and effector functions rely on a delicate balance between activating and inhibitory cell surface ligand-receptor interactions independent of prior sensitization and MHC restriction [42, 43]. We investigated the expression of NK cell-interacting molecules on tumor cell surfaces co-cultured with CAFs in 2D and 3D formats. Notably, tumor cell co-culture with CAFs in 3D led to the downregulation of pro-tumor ligands (HLA-A, B, C, and E) in cancer cells compared to 2D co-culture (Fig 4H). On the other hand, expression of a pro-inflammatory molecule TNFRSF21 is considerably high in both 2D and 3D cultures. Interestingly, the expression levels of IL-15 and IL-18, which are cytokines critical for NK cell maturation, proliferation, and antitumor function, were higher in the 2D co-culture (Fig 4I). Unexpectedly, cytokines, including transforming growth factor TGFb, IL-1b, and IL-10, which can impede NK cell functions, were also upregulated in the 2D co-culture compared to the 3D co-culture.

## CAFs grown in 3D induce monocyte transendothelial migration

Previous studies have highlighted the role of CAFs in modulating immune cell behavior [44, 45]. Specifically, CAFs have been implicated in the recruitment and transendothelial migration of monocytes from circulating vasculature to tumor sites through the secretion of chemokines (including CCL2) and induction of chemotactic gradients [37].

We investigated how conditioned media (CM) derived from CAFs cultured in both 2D and 3D environments affected monocyte transendothelial migration. Conditioned media that were collected from primary PDAC CAFs cultured in both 2D and 3D environments served as a source of cytokines. Human umbilical vein endothelial cells (HUVECs) were seeded onto the top chamber of a transwell and allowed to adhere overnight, forming a confluent monolayer of vascular endothelium. CD14+ monocytes isolated from healthy human donor leukopaks were stained with CellTracker Green CMFDA dye and subsequently added to the top chamber (Fig 5A). Monocyte migration was monitored using kinetic high-content imaging. The captured images illustrate the increased migration of monocytes from the top (blood vessel) to the bottom (tumor compartment) of the Transwell membrane during incubation (Fig 5B). Quantitation of imaging data showed that conditioned media from 3D CAFs considerably enhanced monocyte transendothelial migration compared to control media (Fig 5C). Conditioned media from 2D CAFs promoted only modest monocyte migration, not significantly more than control media (Fig 5C). These data revealed a secreted chemotactic component from 3D CAFs that contributes to monocyte recruitment and tumor infiltration, which is not secreted by 2D CAFs.

To better understand the secreted factors responsible for monocyte transendothelial migration, we examined the expression levels of key cytokines related to monocyte migration in 2D and 3D CAF cultures. While the transcript levels of CCL2 and MMP2, known enhancers of

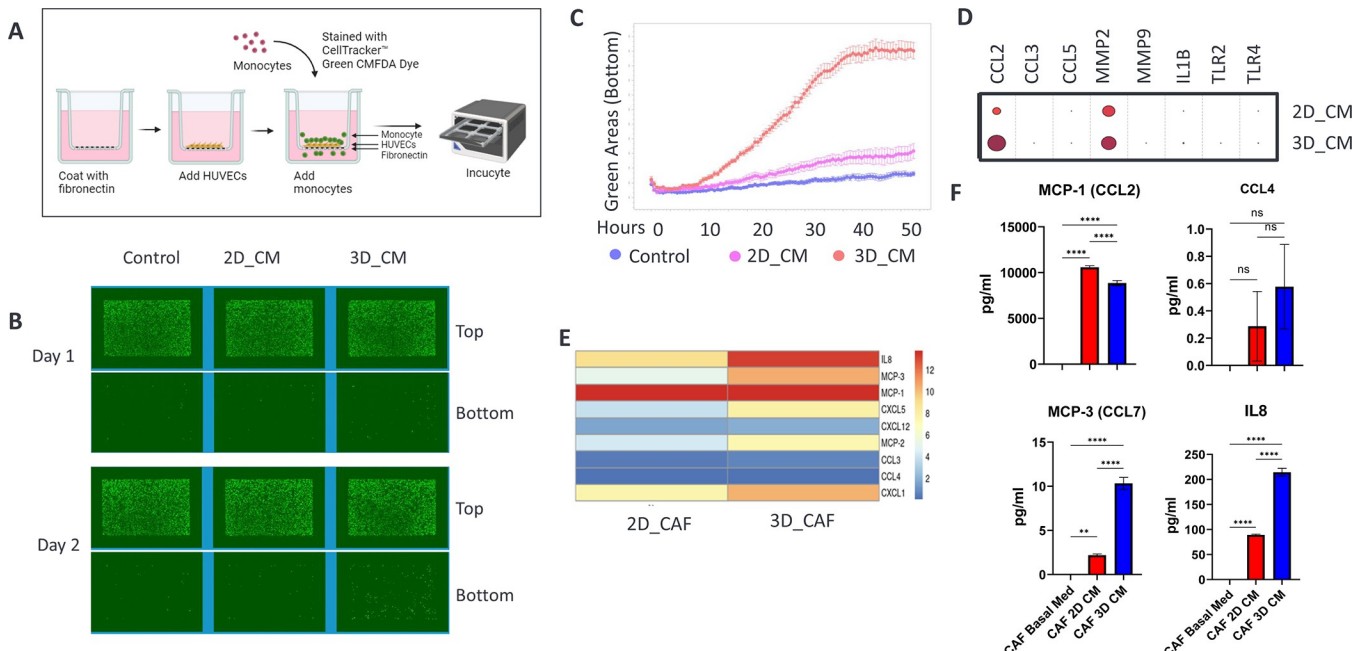

**Fig 5. Conditioned media from 3D CAF culture induced monocyte transendothelial migration.** A) human vascular endothelial cells (HUVECs) were added onto the top chamber of a transwell and incubated overnight to form a confluent monolayer. Then CellTracker green dye pre-stained monocytes were added to the top chamber. Conditioned media from 2D and 3D CAF cultures were used as the chemoattractant and basal culture media as the background control. Monocyte migration was monitored using imaging of the top and bottom membranes by Incucyte. B) Images of top and bottom membranes of transwells taken on day 1 and day 2. C) Kinetic, quantitative data of monocyte migration detected by imaging analysis. D) Transcriptomic gene expression levels of monocyte migration factors from scnRNAseq. E) Relative expression levels of secreted monocyte migration inducing proteins detected in the 2D and 3D CAF conditioned media. F) Quantitation of chemoattractant expression levels in 2D and 2D CAF conditioned media by ELISA.

monocyte migration, were significantly higher in 3D cultured CAFs compared than in 2D cultures (Fig 5D), the protein expression levels of CCL2 (MCP-1) remained comparable in the supernatants from both 2D and 3D CAF monocultures (Fig 5E). However, our results revealed that several other cytokines associated with monocyte migration, such as IL-8, MCP2, MCP-3, CXCL-1, and CXCL-5, were significantly elevated in 3D CAF-conditioned media compared to their 2D counterparts (Fig 5E). Additionally, we performed an enzyme-linked immunosorbent assay (ELISA) to measure the expression of several selected cytokines including IL8, CCL4, CCL2, and CCL7, in the supernatants of 2D and 3D CAFs cultures. We confirmed that the expression levels of CCL2 in 2D and 3D conditioned media were comparable, and validated the increased expression levels of IL-8, CCL4, and CCL7 in 3D CAFs compared with those in 2D CAFs (Fig 5F). These data revealed a clinically relevant chemotactic mechanism [46] that induces monocyte transendothelial migration into tumors, which can only be reproduced using 3D CAF models.

## CAFs grown in 3D induce macrophage differentiation into the protumorigenic M2 subtype

Tumor-associated macrophages (TAMs) are strategically positioned near CAF-populated regions within the tumor stroma, where they exert protumorigenic functions [47]. These immune cells are crudely classified into two distinct subsets, M1 and M2 (where M1 is anti-tumorigenic and M2 is protumorigenic), each is activated by specific polarizing cytokines and plays distinct roles in cancer progression [48–50]. Given the propensity of 3D CAFs to induce

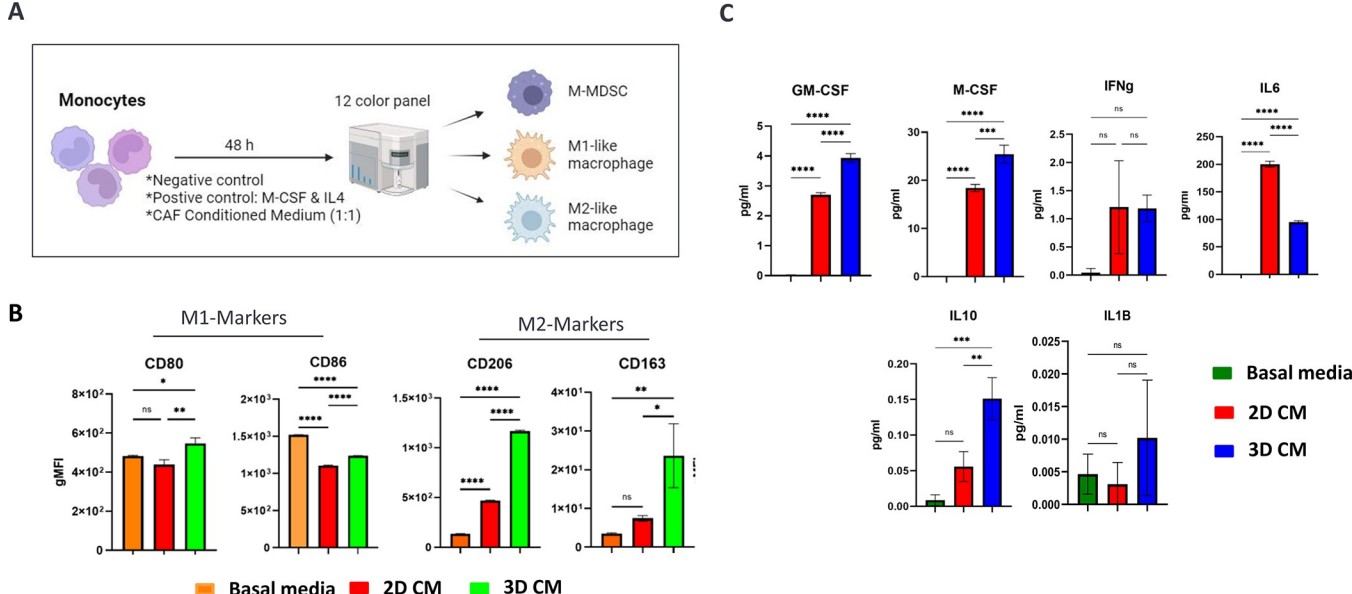

**Fig 6. Conditioned media from 3D CAF cultures induced M2-like macrophage polarization.** A) Monocytes were incubated with base culture media or conditioned media from 2D and 3D CAF cultures for 48 hrs. Biomarkers for macrophages were detected by muti-color FACS analysis. B) Expression levels of biomarkers for M1 and M2 macrophages induced by 2D and 3D conditioned media. C) Monocyte differentiation-inducing cytokines in 2D and 2D CAF conditioned media detected by ELISA.

monocyte transendothelial migration, we next investigated how they would affect subsequent monocyte differentiation and macrophage polarization once infiltrated into the TME.

We cultured monocytes in conditioned media derived from both 2D and 3D CAF cultures for 48 h and then used multicolor flow cytometry analysis to phenotype polarized macrophages based on distinct cell surface marker expression (Fig 6A). Monocytes grown in the presence of 3D CAF-conditioned media demonstrated elevated cell surface expression levels of CD206 and CD163, which are canonical protumoral markers of M2 TAMs, compared with those grown in basal media and 2D CAF-conditioned media (Fig 6B).

CAF-secreted cytokines play critical roles in monocyte recruitment and transformation into M2 macrophages [51, 52]. Specifically, GM-CSF, M-CSF, IL-4, IL-13, IL-8, IL-10, TGF-β, Il-1b, and CCL2 have been shown to promote M2 polarization [45]. Conversely, TNF-α, IFN-γ, TGF-β, VEGF, IL-6, and CXCL12 skew macrophage differentiation toward the M1 phenotype [53]. We quantified the expression levels of several key cytokines in the conditioned media of the 2D and 3D CAF cultures using ELISA. Notably, the levels of M2-inducing cytokines, such as IL-10, Il-1b, GM-CSF, and M-CSF, were significantly higher in 3D CAF-conditioned media than in 2D media (Fig 6C). Conversely, IL-6, an M1 inducing cytokine, was elevated in 2D CAF-conditioned media compared to 3D conditioned media (Fig 6C). Collectively, these observations indicate that CAFs grown in a 3D tumor spheroid format create chemotactic gradients that facilitate monocyte migration from the tumor vasculature and the subsequent polarization into protumorigenic M2 macrophages.

## Discussion

Over 90% of anti-cancer drug candidates have translational failures in clinical trials [54]. A leading cause of this failure can be attributed to poorly-predictive preclinical tumor models that are currently standardized in drug discovery and development [55]. Important tumor stromal components, often overlooked in preclinical models, must be addressed to ensure that

assessment of drug efficacies and their mechanisms of action have clinical translatability [56]. 3D tumor spheroid cultures have proven to be robust surrogates for clinical tumor pathologies, and are recognized for their ability to capture oncological disease states better than conventional 2D tumor formats [57, 58].

CAFs have received increasing attention over the past decade because of their critical roles in tumor initiation and progression in the TME. CAFs play pivotal roles in localized tumor immunosuppression through the transmission of immune-modulating cytokines and soluble factors within the TME [59]. Indeed, CAF-targeting therapies have proven to be synergistic complements to immune checkpoint inhibitors and have provided considerable clinical benefits for patients with CAF-rich tumors [12, 60]. Consequently, the number of preclinical studies aimed at restoring the anti-cancer immune response through CAF-modulating therapies has increased dramatically in recent years [61–63]. Numerous studies have highlighted the importance of the 3D stromal microenvironment in the development of multicellular in vitro cancer models [64, 65]. In this study, using highly sensitive single-nucleus transcriptomic profiling and secretome analyses, we demonstrated that in vitro culture of patient-derived human primary CAFs in 2D and 3D cultures results in different pathological subtypes of CAFs. The identification of the unique transcriptional signatures of myCAFs and iCAFs in 2D and 3D cultures, respectively, suggests that CAF activation is highly sensitive and heavily dependent on dimensional growth and its association with neoplastic cells. While the myCAF subtype is involved in tumor scaffolding and localizes to periglandular tumor regions, iCAFs are associated with more distal TME components, such as vasculature and tertiary lymphoid structures [9, 58], and thus represent a more relevant pathological state for interrogating CAF-immune cell interactions. Interestingly, while CAFs in 3D monocultures display iCAF signatures, CAFs in 3D co-cultures with cancer cells specifically lose their iCAF features [66]. Our data complement previous reports that CAFs are dynamic and can assume different phenotypes based on their spatial and biochemical niches within the PDAC microenvironment [9].

Non-cancerous stromal cells within the tumor microenvironment play pivotal roles in remodeling the ECM, which is a functionally active component of tumors that plays crucial roles in mechanical support, modulation of the microenvironment, and immune cell signaling [67]. Tumor and stromal cells commonly express increased levels of MMPs to remodel the TME for a variety of downstream pathological processes [68–70]. Here, we observed that CAFs grown in 3D cultures upregulated a variety of MMPs such as MMP-1, MMP2, MMP3, and MMP9, which are major players in tumor angiogenesis, tissue remodeling, repair, and maintenance [69, 71, 72]. CAF-derived MMP2 has been strongly implicated in CAF infiltration [72] and is a prerequisite for neoangiogenesis [71]. Interestingly, MMP9, a protease crucial for tumor cell invasion into neighboring tissues and distal metastasis [40], is expressed in both CAFs and cancer cells but only in 3D co-cultures, highlighting the importance of CAF/cancer cell crosstalk for MMP9 secretion. Our results suggest that CAFs cultured in 3D co-cultured tumor spheroids recapitulate cell adhesion and ECM-modifying enzyme secretion, which is consistent with the ECM-mediated cancer-CAF crosstalk found in patient tumors.

We assessed and compared the effects of CAFs on cancer cell growth under 2D and 3D culture conditions, and demonstrated that CAFs in 3D tumor spheroid co-culture with cancer cells promoted tumor cell growth, which is consistent with previous reports [73, 74]. CAFs secrete several growth factors responsible for their pro-proliferative effects on cancer cells. In this study, we observed altered expression of key protumorigenic cytokines and chemokines in 3D cultures compared to 2D cultures, based on secretome profiling.

CAFs play an important role in regulating the antitumor activity of tumor-infiltrating immune cells, including innate and adaptive immune cells in the TME [75]. NK cells are critical regulators of tumor immunosurveillance and lysis of malignant cells, and the manipulation

of NK cell function has been the focus of immunotherapy protocols in cases of resistance to chemotherapy or immune checkpoint inhibitor drugs [76, 77]. Here, we demonstrated that NK cells efficiently killed cancer cells in 2D cancer cell monoculture cultures and co-cultures with CAFs but demonstrated depleted anti-tumor function in a more physiological 3D tumor spheroid model. Several studies have indicated that CAFs exert inhibitory effects on NK cells through multiple processes, including NK receptor activation and cytokine production, either directly or indirectly [37, 78, 79]. CAFs can also indirectly restrict the activity and function of NK cells by modulating the expression of their activating receptor ligands in tumor cells [80]. However, in our study, NK cell death was not associated with the expression of NK inhibitory signals (MHC-I and MHC-II), as tumor cells from 2D co-cultures demonstrated enhanced MHC expression. Similarly, several studies have shown that CAF-secreted TGF-β significantly inhibited the activation and cytotoxic activity of NK cells [81], though in our 2D cultures TGF-β was significantly elevated. The reduced sensitivity of cancer cells to NK cell killing in 3D cultures may be due to the physical barriers of the 3D spheroid architecture, although more granular studies are required to understand these potential mechanisms.

Tumor-associated macrophages (TAMs) are the most prominent immune cells in the vicinity of CAF-populated areas, suggesting important functional interactions between these two cell types [47]. Indeed, the interaction between CAFs and TAMs is scored the strongest within the network of interactions between cell types in human breast cancer TMEs [59]. Our data demonstrated that conditioned media from 3D CAFs facilitated CD14+ monocyte migration and enhanced the polarization of macrophages to the M2 protumorigenic phenotype, consistent with previous reports on CAF/macrophage clinical interactions [48]. Mace et al. documented the central role of CAF-derived macrophage colony-stimulating factor 1 (M-CSF1), IL-6, and CCL2 in monocyte recruitment and increased M2/M1 macrophage ratio in pancreatic cancer [82]. Our 3D models reproduced these findings, suggesting that the factors secreted from CAFs grown as tumor spheroids mimic the clinical mechanisms of monocyte extravasation into the TME and their differentiation into protumorigenic TAM subtypes.

In summary, we compared different ex vivo experimental strategies to mimic clinical CAF biology within a complex human tumor microenvironment. Although both inflammatory and myofibroblastic CAF subtypes can be generated using either 2D or 3D culture formats, 3D tumor spheroid growth in the presence of neoplastic PDAC cells yields the most immuno-relevant CAF. Collectively, our data from the transcriptomic and proteomic profiles of tumor spheroid PDAC CAFs suggest that 3D models yield more immunosuppressive TMEs, manifested through both NK cell and monocyte/macrophage antitumor suppression. As these phenotypes more closely resemble clinical tumors with respect to immune cell presence and function, we propose the integration of 3D TME spheroids into standardized preclinical immuno-oncology pipelines to improve candidate triage and prediction of human therapeutic efficacy.

## Supporting information

**S1 Fig. Quality control (QC) for single-nucleus RNA sequencing (snRNAseq) data analysis.**
(A) Violin plots displaying the QC metrics of the snRNA-seq libraries used in this study after quality control filtering. The distribution of mRNA molecular counts, gene counts, and the percentage of mitochondrial mRNAs within individual libraries are illustrated in the left, middle, and right panels, respectively. (B) Two-dimensional UMAP visualization displays the clustering of 16,235 single-nuclei transcriptomes produced in this study, colored according to the cell culture conditions. Abbreviations: Mono, monoculture of cancer-associated fibroblasts (CAFs); Co, co-culture of CAFs with BxPC3 pancreatic cancer cells; 2D, two-dimensional

monolayer culture; 3D, three-dimensional spheroid cell culture.
(PPTX)

**S1 Table. The panel of macrophage-specific markers for flow cytometry analysis.**
(DOCX)

**S2 Table. IgG controls for flow cytometry analysis.**
(DOCX)

## Acknowledgments

This study was supported by Takeda Development Center of the Americas. We thank Xiang-jun Tian, Shreya Shirodkar, and Charlotte Darby at Takeda Development Center Americas, Inc. for data analysis and members of Core Biology at Takeda Development Center Americas, Inc. for their meaningful discussions and technical support. Figs 1A, 3A, 4A, 5A and 6A were created using the BioRender software.

## Author Contributions

**Conceptualization:** Jian-Ping Yang, Tsubasa Shiraishi, Nicole Aiello, Michael Shaw, Toshihiro Nakamura, Akiko Abiru, Narender R. Gavva, Shane R. Horman.

**Data curation:** Masashi Yamaji, Thang Pham, Han Do.

**Formal analysis:** Masashi Yamaji, Thang Pham, Han Do.

**Funding acquisition:** Narender R. Gavva.

**Investigation:** Jian-Ping Yang, Nikhil Nitin Kulkarni.

**Methodology:** Jian-Ping Yang, Nikhil Nitin Kulkarni.

**Project administration:** Jian-Ping Yang, Shane R. Horman.

**Supervision:** Jian-Ping Yang, Shane R. Horman.

**Validation:** Jian-Ping Yang, Nikhil Nitin Kulkarni.

**Visualization:** Jian-Ping Yang.

**Writing – original draft:** Jian-Ping Yang, Shane R. Horman.

**Writing – review & editing:** Jian-Ping Yang, Tsubasa Shiraishi, Shane R. Horman.

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
