## [Decision Letter · Decision Letter 0]

18 Sep 2024

PONE-D-24-36264Unveiling Immune Cell Response Disparities in Human Primary Cancer-Associated Fibroblasts between Two- and Three-Dimensional CulturesPLOS ONE

Dear Dr. Yang,

Thank you for submitting your manuscript to PLOS ONE, which has been evaluated by two independent reviewers. Both of the referees are interested in the presented study, but also raised some concerns that should be addressed, including revising the introduction and discussion sections, and addtional experiments. Therefore, we invite you to submit a revised version of the manuscript that addresses the points raised during the review process.

We look forward to receiving your revised manuscript.

Kind regards,

Zhiming Li, Ph.D.

Academic Editor

PLOS ONE

Reviewers' comments:

Reviewer's Responses to Questions

**Comments to the Author**

1. Is the manuscript technically sound, and do the data support the conclusions?

Reviewer #1: Yes

Reviewer #2: Yes

2. Has the statistical analysis been performed appropriately and rigorously? 

Reviewer #1: Yes

Reviewer #2: Yes

3. Have the authors made all data underlying the findings in their manuscript fully available?

Reviewer #1: Yes

Reviewer #2: Yes

4. Is the manuscript presented in an intelligible fashion and written in standard English?

Reviewer #1: Yes

Reviewer #2: Yes

5. Review Comments to the Author

Reviewer #1: Yang et al has tried to demonstrate the disparities in human primary cancer associated fibroblast by using 2D and 3D culture. The study design is good but there are some lacks in proper introduction or rational for the study. Authors are suggested to revise the manuscript after incorporated following comments.

1. Studies conducted with 2D and 3D culture of fibroblast cells have been performed using various cancer models, it has been known for a ling time. Authors should highlight the strength of their study over previously published studies. Authors should mention these studies in the introduction section, At line number 82 which can give a broad idea for the readers.

a. PMID: 24124550

b. PMID: 34102862

2. The table 1 and 2 can be added into the supplementary data.

3. The lines 258-262 of the result section can be added in the method/introductory section.

4. Authors have focused the co-culture experiment using pancreatic CAF cells, is there any reason to choose these cells? If yes, then authors are suggested to mention the rational in the introduction section with proper citations.

5. The figure 4H shows the high expression of TNFRSF21 in both 2D and 3D culture which is a pro-inflammatory molecule and enhances the NF-kBeta signaling. What does authors comment on this particular result?

6. Previously conducted studies have also addressed the role of M1 and M2 macrophages and their activation, authors are suggested to cite the following publications at line 415.

a. PMID: 36769180

b. PMID: 38461312

7. Authors should briefly highlight the pros and cons of fibroblast 2D and 3D culture from previous conducted cancer model studies in discussion section.

Reviewer #2: This study examines how 2D and 3D culturing conditions affect cancer-associated fibroblasts (CAFs) in a tumor microenvironment (TME). In 2D cultures, CAFs displayed a myofibroblast (myCAF) subtype, while in 3D tumor spheroids, they shifted to a more inflammatory (iCAF) state. Using RNA sequencing and proteomics, researchers found that 3D CAFs enhanced tumor growth and shielded cancer cells from natural killer (NK) cell attacks. Additionally, 3D CAF-secreted proteins promoted monocyte migration and differentiation into immunosuppressive macrophages. The findings suggest that 3D TME models offer a more clinically relevant system for drug discovery in cancer treatment. To make this manuscript more informative, please consider to perform following experiments.

For example, characteristic changes in CAFs in 3D culture need to be assessed in in vivo setting, using PDAC model mouse. Cytokine production, immune cell infiltration and their characteristics must be evaluated. In addition, authors are encouraged to confirm these findings also in human PDAC samples. For 3D culture, authors should use adequate scaffold such as collagen and other ECM to recapitulate cancer stroma. These settings will further reinforce authors’ claim. Above mentioned experiments seem indispensable to make this manuscript suitable for publication. Please consider to perform these experiments accordingly.

6. PLOS authors have the option to publish the peer review history of their article (what does this mean?). If published, this will include your full peer review and any attached files.

Reviewer #1: **Yes: **NAMRATA ANAND

Reviewer #2: **Yes: **Shin Hamada

---

## [Author Response · Author response to Decision Letter 0]

27 Oct 2024

We are grateful for your and the reviewers’ comments, and the positive evaluation of our work. The followings are our response to the reviewers’ comments: 

Reviewer #1: Yang et al has tried to demonstrate the disparities in human primary cancer associated fibroblast by using 2D and 3D culture. The study design is good but there are some lacks in proper introduction or rational for the study. Authors are suggested to revise the manuscript after incorporated following comments.

1. Studies conducted with 2D and 3D culture of fibroblast cells have been performed using various cancer models, it has been known for a ling time. Authors should highlight the strength of their study over previously published studies. Authors should mention these studies in the introduction section, At line number 82 which can give a broad idea for the readers.

a. PMID: 24124550

b. PMID: 34102862

We thank the reviewer for the additional references and have cited the two papers at line number 82. 

2. The table 1 and 2 can be added into the supplementary data.

We agree with the reviewer and table 1 and 2 have been moved to the supplementary data.

3. The lines 258-262 of the result section can be added in the method/introductory section.

We thank the reviewer for this suggestion but respectfully disagree and feel that these statements are best left in the results section as the first two sentences are a transition into why we used the ensuing methodology.

4. Authors have focused the co-culture experiment using pancreatic CAF cells, is there any reason to choose these cells? If yes, then authors are suggested to mention the rational in the introduction section with proper citations.

We thank the reviewer for pointing out this important point of the research so we have added the following statement:“The microenvironment of pancreatic ductal adenocarcinoma (PDAC) is distinguished by a dense, fibrotic stroma, with cancer-associated fibroblasts (CAFs) serving as pivotal contributors” into line 259 as the rational for this study and the corresponding reference PMID: 36358721 is cited accordingly.

5. The figure 4H shows the high expression of TNFRSF21 in both 2D and 3D culture which is a pro-inflammatory molecule and enhances the NF-kBeta signaling. What does authors comment on this particular result?

The authors thank the reviewer for pointing out this interesting observation. We added “On the other hand, the expression of a pro-inflammatory molecule TNFRSF21 are considerable high in both 2D and 3D culture.” in line 368, although the indication may require further investigation. 

6. Previously conducted studies have also addressed the role of M1 and M2 macrophages and their activation, authors are suggested to cite the following publications at line 415.

a. PMID: 36769180

b. PMID: 38461312

We thank the reviewer for pointing out these two important references and have cited them at line 415. 

7. Authors should briefly highlight the pros and cons of fibroblast 2D and 3D culture from previous conducted cancer model studies in discussion section.

We thank the reviewer of pointing out this very important point that requires further clarification and referencing so we have added “Numerous studies have highlighted the importance of the 3D stromal microenvironment in the development of multiple cell type in vitro cancer models.” in the discussion section of line 458 and added two supporting references.

Reviewer #2: This study examines how 2D and 3D culturing conditions affect cancer-associated fibroblasts (CAFs) in a tumor microenvironment (TME). In 2D cultures, CAFs displayed a myofibroblast (myCAF) subtype, while in 3D tumor spheroids, they shifted to a more inflammatory (iCAF) state. Using RNA sequencing and proteomics, researchers found that 3D CAFs enhanced tumor growth and shielded cancer cells from natural killer (NK) cell attacks. Additionally, 3D CAF-secreted proteins promoted monocyte migration and differentiation into immunosuppressive macrophages. The findings suggest that 3D TME models offer a more clinically relevant system for drug discovery in cancer treatment. To make this manuscript more informative, please consider to perform following experiments.

For example, characteristic changes in CAFs in 3D culture need to be assessed in in vivo setting, using PDAC model mouse. Cytokine production, immune cell infiltration and their characteristics must be evaluated. In addition, authors are encouraged to confirm these findings also in human PDAC samples. For 3D culture, authors should use adequate scaffold such as collagen and other ECM to recapitulate cancer stroma. These settings will further reinforce authors’ claim. Above mentioned experiments seem indispensable to make this manuscript suitable for publication. Please consider to perform these experiments accordingly.

The authors thank the reviewer for pointing out two important points of our research and for the suggested experiments to address those points.

Our current manuscript focuses on comparison of two common in vitro methods for recapitulating human CAF tumor biology. The models proposed here are engineered to mimic human-specific tumor biology, which may not be represented in murine subcutaneous tumor models. Therefore, validating our findings through mouse models would not support our point of clinical translatability. That is the reason we have chosen to use clinically-derived snRNAseq data sets to support our findings. Given the new FDA 2.0 act which does not require rodent models for advancing clinical therapeutics, we feel the human tumor models proposed here are particularly relevant.

Regarding the use of collagen or other type of ECM in our 3D models, the authors appreciate the suggestion. Typically, exogeneous ECM is incorporated into 3D tumor models that require these reagents to “stick” the various cell types together. Given that BxPC3 cells are a secretory ductal cell type, our 3D PDAC models did not require addition of exogeneous ECM. Further, snRNAseq data from the CAF cells show important ECM components are additionally secreted from the primary CAFs as well, consistent with clinical CAF function.

Regarding the use of primary PDAC samples, we used BxPC3 human PDAC cells in our study due to their desirable oncogenic mutational profile of Braf*, TP53* and ease of 3D culture. We were unable to locate a primary PDAC sample with a similar mutational profile. Additionally, we felt the incorporation of both primary human CAFs and PBMCs into our models offered a translational nature to our tumor model irrespective of tumor cell origin.

Having said that, we agree with the reviewer that future studies incorporating primary patient-derived PDAC tumor samples may reveal novel tumor cell – CAF interactions.

6. PLOS authors have the option to publish the peer review history of their article (what does this mean?). If published, this will include your full peer review and any attached files.

Yes, the authors wish to publish the peer review history of our article.

We again thank the reviewers for their thoughtful edits and suggestions and have revised and modified the text according to reviewers’critiques. We believe these changes have improved the manuscript considerably and we hope it can be published without delay.

Sincerely，

Jian-Ping Yang, PhD

---

## [Decision Letter · Decision Letter 1]

7 Nov 2024

Unveiling Immune Cell Response Disparities in Human Primary Cancer-Associated Fibroblasts between Two- and Three-Dimensional Cultures

PONE-D-24-36264R1

Dear Dr. Yang,

Thank you for the efforts in revising your manuscript. We’re pleased to inform you that your manuscript has been judged scientifically suitable for publication and will be formally accepted for publication once it meets all outstanding technical requirements.

Kind regards,

Zhiming Li, Ph.D.

Academic Editor

PLOS ONE

Additional Editor Comments (optional):

Reviewers' comments:

Reviewer's Responses to Questions

**Comments to the Author**

1. If the authors have adequately addressed your comments raised in a previous round of review and you feel that this manuscript is now acceptable for publication, you may indicate that here to bypass the “Comments to the Author” section, enter your conflict of interest statement in the “Confidential to Editor” section, and submit your "Accept" recommendation.

Reviewer #1: All comments have been addressed

2. Is the manuscript technically sound, and do the data support the conclusions?

Reviewer #1: Yes

3. Has the statistical analysis been performed appropriately and rigorously? 

Reviewer #1: Yes

4. Have the authors made all data underlying the findings in their manuscript fully available?

Reviewer #1: Yes

5. Is the manuscript presented in an intelligible fashion and written in standard English?

Reviewer #1: Yes

6. Review Comments to the Author

Reviewer #1: (No Response)

7. PLOS authors have the option to publish the peer review history of their article (what does this mean?). If published, this will include your full peer review and any attached files.

Reviewer #1: **Yes: **NAMRATA ANAND

---

## [Editor Report · Acceptance letter]

8 Dec 2024

PONE-D-24-36264R1 

PLOS ONE

Dear Dr. Yang, 

I'm pleased to inform you that your manuscript has been deemed suitable for publication in PLOS ONE. Congratulations! Your manuscript is now being handed over to our production team.

Kind regards, 

on behalf of

Dr. Zhiming Li 

Academic Editor

PLOS ONE